# Antibiotics utilization patterns among tertiary care hospitals in Ethiopia

**Abebe Kebede**[1]*, **Kifle Woldemichael**[2], **Dereje Kebede**[1‡], **Sileshi Dubale**[3‡],
**Wendossen Shewarega**[4‡], **Alemseged Beyene Berha**[5‡], **Sultan Suleman**[1]

1 Jimma University Laboratory of Drug Quality (JuLaDQ), School of Pharmacy, Institute of Health, Jimma University, Jimma, Ethiopia, 2 Department of Epidemiology, Institute of Health, Jimma University, Jimma, Ethiopia, 3 School of Pharmacy, College of Health Sciences, Mattu University, Mattu, Ethiopia, 4 Federal Ministry of Health, Addis Ababa, Ethiopia, 5 Department of Pharmacology and Clinical Pharmacy, School of Pharmacy, College of Health Sciences, Addis Ababa University, Addis Ababa, Ethiopia

☯ These authors contributed equally to this work.
‡ DK, SD, WS and ABB also contributed equally to this work.
* abekebe30@gmail.com

## Abstract

### Background

Antibiotics are among the most used medicines globally, but antimicrobial resistance (AMR) threatens their effectiveness. The greatest mortality burden associated with AMR is in sub-Saharan Africa. However, antimicrobial prescribing practice and stewardship remain challenges in the African regions. Thus, this study aimed to evaluate and compare antibiotic utilization patterns in tertiary hospitals in Ethiopia during 2022.

### Method

A retrospective cross-sectional study was conducted in adult wards of five public tertiary care hospitals in Ethiopia with a total of 3,283 beds. Data were retrieved from 807 randomly selected patient records using the online Kobo tool. Analysis utilized the World Health Organization (WHO) Anatomical Therapeutic Chemical (ATC) classification system and the Defined Daily Dose (DDD) (ATC/DDD) method. The result was presented using tables, charts, and text.

### Results

A total of 2,718 drugs were prescribed to 807 patients with an average of 3.4 drugs per patient (range 1–7) during a total of 8638 bed-days. Of the total drugs prescribed, antibiotics account for 1,035 (38%), with an average of 1.3 (1,035/807) antibiotics per patient. Five hundred fifty-six (69%) patients were prescribed at least one antibiotic. The overall antibiotic consumption was 108 DDD/100 bed-days, 37.5/100 bed-days for hospital-acquired infections, and 32.8/100 bed-days for community-acquired infections. The rest were for prophylaxis purposes. The majority of

**Data availability statement:** All relevant data are within the manuscript and its Supporting information files.

**Funding:** The author(s) received no specific funding for this work.

**Competing interests:** The authors have declared that no competing interests exist.

antibiotics were prescribed in medical and surgical wards; 34.9/100 and 27.5/100 bed-days, respectively. The most prescribed antibiotics were ceftriaxone, metronidazole, and ceftazidime. Overall, the AWaRe "Watch" group antibiotics use occurred in 74% (73 DDD/100 bed-days) of total antibiotic consumption, which was higher than the WHO recommendation (at least 60% of total antibiotic use should be from the "Access" group, not the "Watch" group).

## Conclusion

Antibiotic use was high in Ethiopian tertiary hospitals, with most patients receiving antibiotics, mainly from the WHO "Watch" group, contrary to guidelines. Three classes (cephalosporins, imidazoles, and glycopeptides) made up the majority of prescriptions, mostly for hospital-acquired infections. Urgent interventions and strengthened antimicrobial stewardship are needed to address inappropriate use and combat resistance.

## Introduction

The introduction of antimicrobials inhibiting the growth and replication of bacteria or outright killing them [1] was a key breakthrough in human history. As predicted soon after their discovery, resistance has become a major threat [2], accelerated by uncontrolled and inappropriate consumption [3]. The consequent spread of antimicrobial resistance (AMR) results in increased morbidity, mortality, and treatment costs [4]. Hence, reducing unnecessary and inappropriate antibiotic use, which can include overuse, misuse, underuse, or abuse of antibiotics, is a public health priority [5,6]. The amount of antibiotics prescribed, the number of patients treated with the antibiotics, and the proportion of patients on antibiotics in a hospital are three important factors contributing to AMR [7].

Antibiotic consumption is defined as the quantity of antibiotics used in a specific setting during a specific period. For global reporting purposes, national estimates of antibiotic consumption are typically reported for a period of a full calendar year (January to December) [8,9]. According to the World Health Organization (WHO) report, global consumption of antibiotics increased from 8.2 to 13.6 Defined Daily Dose (DDD) per 1000 inhabitants per day from the year 2000–2015, accounting for a 39.7% rise; and between 2016 and 2018 from 14.4 to 64.4 DDD per 1000 inhabitants per day [9].

In Ethiopia, community-based consumption of systemic antibiotics increased by 16.4% from 2016 to 2020; while the average growth rate per year between 2016 and 2020 was 3.3% [1]. A study conducted at the largest tertiary care hospitals in Ethiopia revealed that the mean antibiotic consumption during 1 year in three wards was 81.6 DDD/100 Bed-Days (BD) [10]. A study conducted in Jimma University Medical Center (JUMC) reported that the majority of antibiotics were prescribed from the "Watch" group, 51.7%, while 48.3% were from the "Access" group. Ceftriaxone was the most commonly prescribed antibiotic [11].

Determining the amount and pattern, as well as class of antibiotics, is very important for setting national targets and guiding clinical efforts and policies that promote improved antibiotic use [7,12]. The amount and trend of antibiotic consumption in low- and middle-income countries, including Ethiopia, remains inadequately studied.

Despite limited studies on utilization patterns of antibiotics in Ethiopia, most research has been conducted at single centers, and data on the overall pattern across multiple hospitals remain scarce. Understanding the current status of antibiotic utilization patterns is essential for informing proactive infection control measures, guiding policy development, and optimizing resource allocation. Our findings provide valuable insights for healthcare professionals, hospital administrators, and policymakers, offering evidence-based recommendations for optimizing antimicrobial use strategies at both institutional and national levels. Thus, the objective of this study was to evaluate and compare antibiotic utilization patterns in five tertiary care hospitals in Ethiopia.

## Materials and methods

### Study design and setting

A retrospective cross-sectional study was conducted in adult medical, surgical, obstetrics/gynecology, and intensive care unit (ICU) wards of five public tertiary care hospitals in Ethiopia between January and December 2022. The five hospitals involved in the study were Jimma University Medical Center (JUMC), Wolaita Sodo University Comprehensive Specialized Hospital (WSUCSH), Hawassa University Comprehensive Specialized Hospital (HUCSH), Tikur Anbesa Comprehensive Specialized Hospital (TASH), and St. Paul's Hospital Millennium Medical College (SPHMMC), comprising a total of 3,283 beds, located across three regions and the capital city of Ethiopia.

### Inclusion and exclusion criteria

All adult patients aged 18 years and above admitted to the study wards of tertiary care hospitals, and who were prescribed antibiotics during their hospital stay, were included. All antibiotics prescribed for treatment and prophylaxis were included. Patients with a hospital stay of 3 months or longer, as well as oncology or hematology patients, were excluded.

### Sample size estimation

The sample size was calculated by using the single population proportion formula StatCalc of Epi Info version 7.2.5.0 software package, taking into account the following assumptions: 50% expected frequency of antibiotic consumption, 95% confidence interval (CI), and 5% margin of error. By applying a design effect of 2 and adding 5% to compensate for non-response, the final calculated sample size was found to be 807 patients' medical records.

### Sampling method

The sample size was determined using a multi-stage random sampling method. In the first stage, 30% of all regional states and city administration units in Ethiopia were randomly selected. In the second stage, based on the total number of tertiary care hospitals located in the included regions and city administrations, the number of selected tertiary care hospitals in each region and city administration was determined in a ratio of 3:1 (from every three tertiary care hospitals, one tertiary care hospital was selected from each regional state and city administration selected for the study). Then, the tertiary care hospitals in each region and the city administration were randomly selected. From the region and city administration, with a total number of tertiary care hospitals of less than three, one hospital was selected by the lottery method. Finally, the total sample was allocated to all study tertiary care hospitals and wards within each hospital based on the number of beds shown in the diagram below (Fig 1: diagram of multi-stage random sampling method to select study unit separately attached).

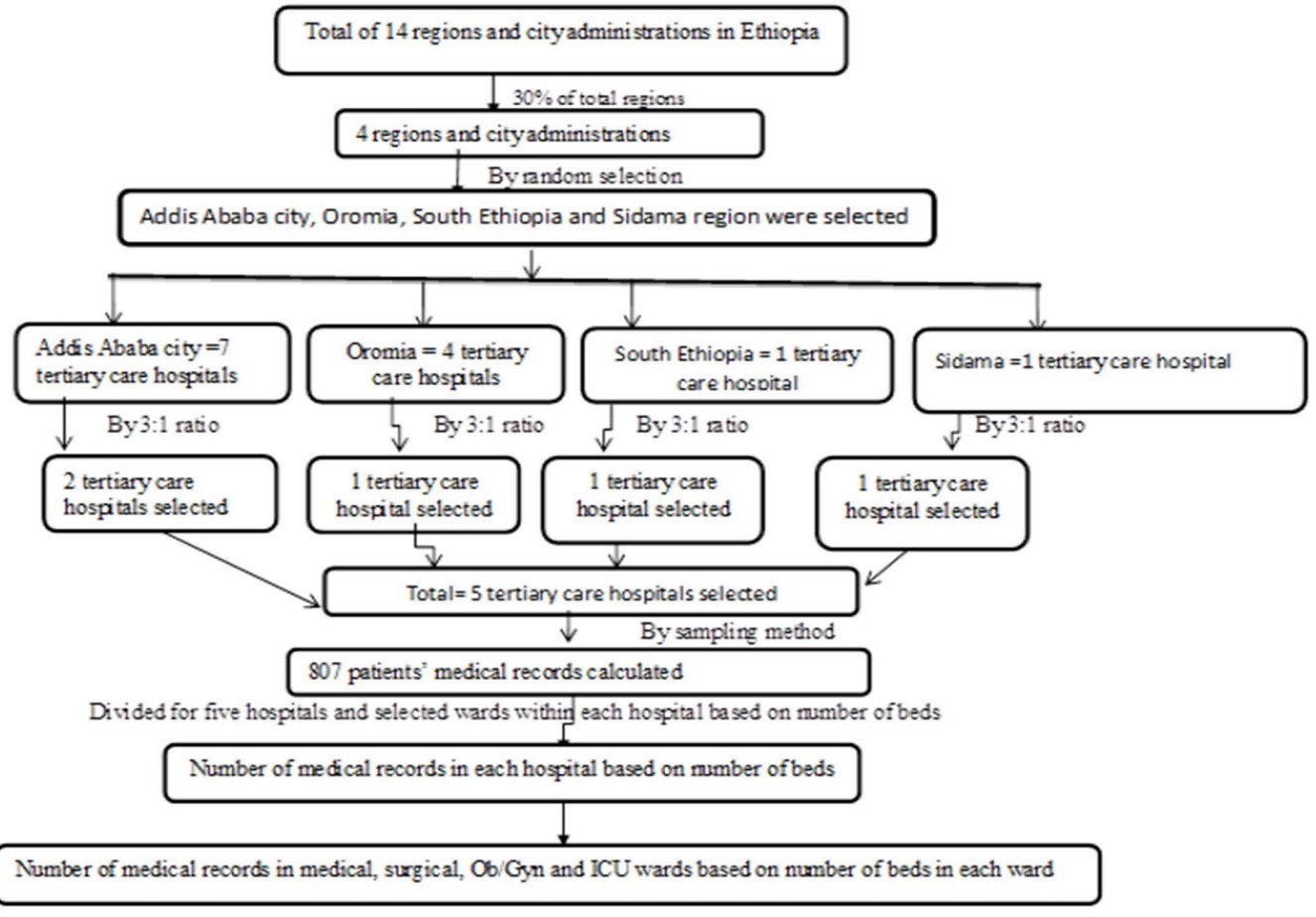

**Fig 1. Diagram of sampling procedure.**

According to the Ethiopian Ministry of Health (MoH) report on health and health-related Indicators (2021/2022), there exist four tertiary care hospitals in Oromia, one in the South Ethiopia, one in the Sidama region, and seven in the Addis Ababa city administration [13]. Accordingly, Jimma University Medical Center (JUMC) from the Oromia, Wolaita Sodo University Comprehensive Specialized Hospital (WSCSH) from the South Ethiopia, Hawassa University Comprehensive Specialized Hospital (HUCSH) from the Sidama regions, and Tikur Anbesa Comprehensive Specialized Hospital (TASH) and St. Paul's Hospital Millennium Medical College (SPHMMC) from the Addis Ababa city administration were the five selected tertiary hospitals for the study. After proportionally allocating 807 patient medical records to the five selected tertiary care hospitals and their respective four wards based on the number of beds in each hospital, a systematic random sampling technique was employed to select patient records from the admission registration books of the medical, surgical, obstetrics/gynecology, and ICU wards. This means that after arranging patients' medical records sequentially based on their registration number, the first record was selected by lottery method from the first five charts, and subsequent records from each ward were selected using a fixed interval to complete the sample. The data from each study hospital and ward were collected at intervals of the ratio of allocated sample size to annual admission in each selected ward within the study hospital until the expected number of patients from each ward was filled.

## Data collection procedure and instruments

The data sources were patients' records, with each record representing one patient. The data collected included demographics (medical record number, age, sex, and ward of admission), clinical characteristics (indication of antibiotic use, length of hospital stay, and outcome), and drug use data (name, dose, strength, frequency, quantity, duration, antibiotic type, class, AWaRe category, and ATC code).

All sampled patients' medical records were selected through a systematic sampling technique from the admission registry of 2022 from each ward, using the patient's medical record number (MRN). The selected patients' medical records were manually retrieved from the central medical record center. The ATC/DDD methodology was used to standardize the data collection and reporting of antibiotic consumption. The Kobo collection tool was completed using the patients' medical records. A standardized data collection form was developed based on the World Health Organization (WHO) criteria for drug use evaluation.

## Data collectors and quality assurance

Data were collected by a total of ten trained health professionals, with two data collectors recruited from each hospital. A one-day training session was conducted to orient them on the study's objective and the use of the Kobo Collect tool for data collection. The data collection tool was pre-tested on 25 randomly selected patient medical records, which were excluded from the final analysis.

## Ethical issues

Before data collection, ethical clearance was obtained from the institutional Ethical Review Board (IRB) of Jimma University, Institute of Health (Ref. number JUIH/IRB/629/23 on date 30/10/2023). The support letter to each study hospital was obtained from Jimma University School of Pharmacy and was submitted to the respective tertiary care hospitals (Ref. No. Pharmacy/School 271/2016 on date 08/11/2023). Lastly, the verbal permission obtained from each hospital was communicated to the central medical registration office of the corresponding hospital. In addition, the confidentiality of personal information was strictly observed.

In this study, we report a retrospective study of medical records in which all data were fully anonymized before we accessed them. We reviewed patients' medical records that were documented while providing medical services and were obtained in hospital medical record centers after the discharge of the patients from hospitals. We randomly selected the required sample size from these medical records by systematic sampling technique using the patient's medical registration number (MRN) in the absence of patients. Consequently, medical records were assessed without patients' participation. Hence, as there were no human participants involved and medical records were selected without awareness of patients, informed consent was not needed.

## Periods of study and data collection

The period of study was January 01/2023, to December 30/2023. The data were collected from April 15/2024, to August 03/2024.

## Data analysis and interpretation

Following data collection, the dataset was validated for completeness, then downloaded into Microsoft Excel for coding and subsequently exported to SPSS Statistics version 21 for analysis. Descriptive statistics were used, including frequency distributions and percentages for categorical variables (e.g., antibiotic types, indications) and mean±standard deviation for continuous variables (e.g., DDDs, bed-days). To quantify antibiotic consumption, the percent contribution of each antibiotic to the total annual antibiotic use was calculated. The measurement unit employed was DDD/100 bed days

(DDD/100 bed-days). Aggregated antibiotic consumption data were organized according to the ATC classification and expressed in two standardized metrics: (1) Defined Daily Doses (DDDs), to quantify total consumption; and (2) DDD/100 bed-days, which adjusts for patient occupancy to facilitate cross-facility comparisons.

## Calculation and measurement of antibiotic consumption

The Anatomical Therapeutic Chemical (ATC)/Defined Daily Dose (DDD) system, endorsed by WHO, was used for the quantitative evaluation and comparison of antibiotic use [14]. The ATC classification system was used to organize drugs by therapeutic and anatomical properties, while the DDD was used to calculate the fixed average daily dose [2].

DDD was calculated using the formula:

$$DDDs = Total\ grams\ of\ antibiotic\ used/DDD\ value\ (grams) \tag{1}$$

Where total grams were derived by summing active ingredients across formulations (tablets or capsules, syrups, injections) and packaging.

The DDD/100 bed-days metric was calculated as:

$$DDD/100\ bed-days = \left( \frac{Total\ DDDs}{Patient\ days} \right) * 100 \tag{2}$$

## Results

The total final sample included 807 patients, yielding a 100% response rate, with each medical record representing one patient. The majority of the study participants were female (n = 425, 53%) and aged 18–27 years. A significant proportion of participants (n = 489, 62%) were from two tertiary care hospitals: JUMC and TASH. More than two-thirds of the participants were admitted to medical and surgical wards (Table 1).

A total of 2,718 drugs were prescribed to 807 patients, with an average of 3.4 drugs per patient (range: 1–7). Of these, 1,035 (38%) were antibiotics. This corresponds to an average of 1.3 antibiotics prescribed per patient (1,035/807) (Table 2).

The majority (n = 556, 69%) of patients were prescribed a single antibiotic, while (n = 254, 31%) received two or more antibiotics during their hospital stay. JUMC recorded the highest antibiotic use, with 182 of 249 (73%) patients being prescribed at least one antibiotic and (n = 103 41%) two or more. In contrast, WSUCSH had the lowest antibiotic consumption, with only 54 of 110 patients (49%) receiving at least one antibiotic and 22 (20%) receiving two or more antibiotics. The average number of antibiotics prescribed per patient varied from 0.7 at WSUCSH to 1.14 at JUMC (Table 3).

Regarding indication for antibiotic use, the majority (70.3 DDD/100BD, 65.3%) of antibiotics were prescribed for therapeutic purposes for hospital-acquired infections (37.5 DDD/100BD, 34.69%) and community-acquired infections (32.8 DDD/100BD, 30.34%).; while the rest were for prophylaxis purposes (Table 4).

## Antibiotic consumption using DDD methodology

In this study, 18 types of antibiotics from 13 classes were prescribed. The overall annual antibiotics consumption was found to be 107.64 DDD/100-BD. The annual consumption of antibiotics showed considerable variations among hospitals. Notably, JUMC (46.19 DDD/100 BD) and TASH (31.74 DDD/100 BD) together accounted for over three-fourths of the total annual antibiotics consumption. Among the antibiotics, ceftriaxone was the most frequently prescribed (41.29 DDD/100 BD), followed by Metronidazole (18.98 DDD/100 BD) and Ceftazidime (11.94 DDD/100 BD) (Table 5).

Of the total annual antibiotic consumption of 108 DDD/100 BD, the medical ward accounted for the highest share (34.92 DDD/100BD), followed by the surgical ward (27.5 DDD/100BD) and the obstetrics/gynecology ward (23.59

**Table 1. The Socio-demographic characteristics of the study participants and study setting in Ethiopia, 2022.**

| Category | Description | # | % |
|---|---|---|---|
| **Sex** | Male | 382 | 47 |
| | Female | 425 | 53 |
| **Age group in years** | 18-27 | 231 | 29 |
| | 28-37 | 197 | 24 |
| | 38-47 | 136 | 17 |
| | 48-57 | 112 | 14 |
| | 58-67 | 76 | 9 |
| | 68-77 | 35 | 4 |
| | >77 | 20 | 3 |
| **Wards of admission** | Medical ward | 294 | 36 |
| | Surgical ward | 261 | 32 |
| | Obstetrics/Gynecology ward | 169 | 21 |
| | Intensive care unit(ICU) | 83 | 10 |
| **Hospitals of admission** | Jimma University Medical Center | 249 | 31 |
| | Tikur Anbessa Specialized Hospital | 249 | 31 |
| | Wolaita Sodo University Specialized Hospital | 110 | 13 |
| | Hawassa University Comprehensive Specialized Hospital | 102 | 14 |
| | St. Paulos Hospital Millennium Medical College | 97 | 12 |

**Table 2. Total number of drugs prescribed and number and ratio of antibiotics from total prescribed drugs.**

| # of patients | Drugs | Total # of drugs prescribed | # of antibiotics prescribed | # of non-antibiotics prescribed | Ratio of antibiotics |
|---|---|---|---|---|---|
| **807** | Drug 1 | 805 | 556 | 249 | 69.07 |
| | Drug 2 | 650 | 242 | 408 | 37.23 |
| | Drug 3 | 565 | 123 | 442 | 21.77 |
| | Drug 4 | 366 | 64 | 302 | 17.49 |
| | Drug 5 | 176 | 29 | 147 | 16.48 |
| | Drug 6 | 94 | 13 | 81 | 13.83 |
| | Drug 7 | 62 | 8 | 54 | 12.90 |
| | **Total** | **2718** | **1035** | **1683** | **38** |

Note: Drug 1, Drug 2, Drug 3, Drug 4, Drug 5, Drug 6, and Drug 7 were prescribed medication lists written/prescribed as 1st,2nd,3rd,4th,5th,6th, and 7th rounds (times) during patients' hospital stay.

DDD/100BD). The intensive care unit (ICU) recorded the lowest annual antibiotic consumption, at 21.64 DDD/100BD (Table 6).

### Distribution of antibiotic classes

Among the 13 different classes of antibiotics prescribed, only three accounted for approximately 80% of all prescriptions. Third-generation cephalosporins were the most frequently used (55%), followed by imidazoles (13.7%) and glycopeptides (10.7%) (Table 7).

### WHO AWaRe classification of prescribed antibiotics

Of the 18 types of antibiotics prescribed, 10 belonged to the WHO "Access" group and 8 to the "Watch" group. Out of the total 1,035 antibiotic prescriptions, 768 (74%) were from the "Watch" group, while the remaining were from the "Access"

**Table 3. Number of antibiotics prescribed per patient and antibiotics exposure rate in adult patients admitted to medical, surgical, obstetrics/gynecology, and ICU wards of tertiary care hospitals in Ethiopia during 2022.**

| Hospitals | Total # of patients | # of Pts prescribed one antibiotic | # of Pts prescribed two antibiotics | #of Pts prescribed three antibiotics | # of Pts prescribed four antibiotics | % of patients prescribed at least one antibiotic | % of patients prescribed at least two antibiotics | The exposure rate of antibiotics in each hospital |
|---|---|---|---|---|---|---|---|---|
| JUMC | 249 | 182 | 74 | 20 | 9 | 73 | 41.4 | 1.2 |
| TASH | 249 | 176 | 64 | 16 | 2 | 70.7 | 32.9 | 1.0 |
| HUCSH | 102 | 69 | 29 | 4 | 0 | 67.7 | 32.4 | 1.0 |
| WSUCSH | 110 | 54 | 19 | 3 | 0 | 49 | 20.0 | 0.7 |
| SPHMMC | 97 | 75 | 12 | 2 | 0 | 77 | 14.4 | 0.9 |
| **Total** | **807** | **556** | **198** | **45** | **11** | **69** | **31.5** | **1.00** |

Note: JUMC Jimma University Medical Center, WSUCSH = Wolaita Sodo University Comprehensive Specialized Hospital, HUCSH = Hawassa University Comprehensive Specialized Hospital, TASH = Tikur Anbesa Comprehensive Specialized Hospital. SPHMMC = St. Paul's Hospital Millennium Medical College.

**Table 4. Antibiotic consumption(DDD/100BD) based on the indication in adult patients admitted in medical, surgical, obstetrics/gynecology, and ICU wards of tertiary care hospitals in Ethiopia during 2022.**

| Drugs | # of antibiotics prescribed | # of antibiotics prescribed for community-acquired infections | # of antibiotics prescribed for hospital-acquired infections | # of antibiotics prescribed for surgical prophylaxis | # of antibiotics prescribed for medical prophylaxis |
|---|---|---|---|---|---|
| Drug 1 | 556 | 256 | 145 | 72 | 83 |
| Drug 2 | 242 | 27 | 95 | 64 | 56 |
| Drug 3 | 123 | 11 | 62 | 25 | 25 |
| Drug 4 | 64 | 9 | 31 | 14 | 10 |
| Drug 5 | 29 | 4 | 17 | 6 | 2 |
| Drug 6 | 13 | 3 | 6 | 2 | 2 |
| Drug 7 | 8 | 4 | 3 | 1 | 0 |
| **Total #** | **1035** | **314** | **359** | **184** | **178** |
| **%** | **100** | **30.34** | **34.68** | **17.78** | **17.20** |
| **DDD/100BD** | **108** | **32.7** | **37.5** | **19.2** | **18.6** |

Note: Drug 1, Drug 2, Drug 3, Drug 4, Drug 5, Drug 6, and Drug 7 indicate types of medications prescribed and written on patients' medical records or prescription changed for patients as 1st,2nd,3rd,4th,5th,6th, and 7th rounds (times) during patients' hospital stay.

group. In terms of Defined Daily Dose (DDD), the consumption of "Watch" group antibiotics was approximately twice that of the "Access" group (Table 8).

## Discussion

In the current study, the utilization patterns of antibiotics have been evaluated and compared using the WHO ATC/DDD methodology at four major wards of five public tertiary care hospitals in Ethiopia. The results of this study showed that among the total prescribed medications, 38% were antibiotics, which exceeded the WHO-recommended standard of 20–26.8% [15]. There was a strong relationship between antibiotic consumption and antibiotic resistance [16]. The excessive exposure to antibiotics results in the emergence and spread of antimicrobial resistance [2], which potentially results in increased morbidity, mortality, and treatment costs [4]. Since every time an antimicrobial drug is used, it diminishes the effectiveness for all users [12]. High rates of resistance against frequently used antibiotics to treat infections, resulting in running out of effective antibiotics to treat common infections [16]. Resistance against antimicrobials, specifically, is an

**Table 5. Antibiotic consumption of adult patients admitted in medical, surgical, obstetrics/gynecology, and ICU wards among tertiary care hospitals in Ethiopia during 2022.**

| Drugs | ATC code | Class of antibiotics | Total DDD | DDD/ 100 BD | JUMC DDD (DDD/ 100 BD) | TASH DDD (DDD/ 100BD) | HUCSH DDD (DDD/ 100 BD) | WSUCSH DDD (DDD/ 100BD) | SPMMC DDD (DDD/ 100BD) |
|---|---|---|---|---|---|---|---|---|---|
| Amoxicillin | J01CA04 | Penicillin | 49 | 0.57 | 49 (0.57) | 0 | 0 | 0 | 0 |
| Amoxicillin + Clavulanic acid | J01CR02 | Beta-lactam-beta-lactamase inhibitors | 178.33 | 2.07 | 47.5(0.6) | 114.6(1.4) | 0 | 16.3(0.2) | 0 |
| Ampicillin | J01CA01 | Penicillin | 183.67 | 2.13 | 6.7(0.1) | 85(0.98) | 10(0.12) | 0 | 82(0.95) |
| Azithromycin | J01FA10 | Macrolides | 70.75 | 0.82 | 15.3(0.2) | 5 (0.06) | 5 (0.06) | 44.5(0.5) | 1(0.01) |
| Benzanthine penicillin | J01CE08 | Penicillin | 0.048 | 0.001 | 0 | 0 | 0.03(0.1) | 0.02(0.01) | 0 |
| Cephalexin | J01DB01 | First-generation cephalosporin | 182.5 | 2.11 | 157.5 (2) | 20 (0.23) | 0 | 5(0.06) | 0 |
| Cefepime | J01DE01 | Fourth-generation cephalosporin | 147 | 1.70 | 0 | 42.25(0.5) | 92.3(1.1) | 3 (0.03) | 9.5(0.11) |
| Ceftazidime | J01DD02 | Third-generation cephalosporin | 1031.5 | 11.94 | 401(4.6) | 373 (4.3) | 78 (0.9) | 65 (0.8) | 114.5(1.3) |
| Ceftriaxone | J01DD04 | Third-generation cephalosporin | 3567 | 41.29 | 1405 (16.3) | 1007 (11.7) | 511.5 (5.92) | 511.5 (6) | 132.5 (1.53) |
| Ciprofloxacin | J01MA02 | Fluoroquinolones | 369.8 | 4.28 | 107(1.3) | 152.4(1.8) | 40.8(0.5) | 15(0.2) | 54.6(0.6) |
| Cloxacillin | J01CF02 | Penicillin | 32 | 0.37 | 20 (2.3) | 7 (0.1) | 0 | 0 | 5(0.06) |
| Sulphamethoxazole + trimethoprim (Cotrimoxazole) | J01EE01 | Sulfonamides and trimethoprim | 38.88 | 0.45 | 0 | 0 | 30.7(0.4) | 0 | 8.16(0.1) |
| Doxycycline | J01AA02 | Tetracycline | 688 | 7.97 | 660(7.64) | 28 (0.32) | 0 | 0 | 0 |
| Gentamicin | J01GB03 | Aminoglycosides | 27.01 | 0.31 | 3.2 (0.15) | 3.3 (0.16) | 0 | 0 | 0 |
| Meropenem | J01DH02 | Carbepenem | 80.25 | 0.93 | 18.3(0.2) | 42 (0.5) | 62(0.72) | 0 | 0 |
| Metronidazole | J01XD01 | Imidazole | 1639.01 | 18.98 | 634.9(7.4 | 615 (7.12) | 211.6 (45) | 124.5(1.4) | 53(0.61) |
| Norfloxacin | J01MA06 | Fluoroquinolones | 28 | 0.32 | 5.6 (0.06) | 11.2(0.13) | 11.2(0.1) | 0 | 0 |
| Vancomycin | J01XA01 | Glycopeptides | 985 | 11.40 | 448.5(5.2 | 226(2.62) | 172.5 (2) | 76 (0.88) | 62(0.72) |
| Total(N,%) | | | 9297.8 | 108.0 | 3989.5 (46.2) | 2742.10 (31.7) | 1183.6 (13.7) | 860.3 (10.0) | 522.3 (6.0) |

urgent problem because antibiotics are a cornerstone of modern medicine, and most medicinal procedures in human and animal health rely on functioning antibiotics [12]. Hence, optimizing the use of antimicrobials is very important and one of the national antimicrobial resistance prevention and containment strategic plans of Ethiopia. Countries, regions, or health care facilities with high antimicrobial use and misuse are associated with a high incidence of resistance. Evidence-based policies, protocols, and regulations that encourage more appropriate and rational use of antimicrobials are key interventions for the containment of antimicrobial resistance [17].

The current study revealed that the rate of patients who were prescribed at least one antibiotic during their hospital stay was 69.09%. This percentage was higher than the prior studies: Debre Tabor, in Ethiopia, 60.6% [18], Adgirat Hospital, North Ethiopia, 44.5% [19], and Axum Comprehensive Specialized Hospital, Northern Ethiopia, in which 52.3% of patients had at least one oral and/or injectable antibiotic prescribed [20]. On the contrary, there was a much higher antibiotic prescription pattern reported from Eritrea, 79% [21], Iran, 73% [22], and Pakistan, 82.3% [23].

**Table 6. Antibiotic Consumption in Adult Wards of Tertiary Care Hospitals in Ethiopia, 2022.**

| Drug | ATC code | Class of antibiotics | DDD | Total DDD/ 100 BD | Medical ward DDD (DDD/100BD) | Surgical ward DDD (DDD/100) | OB/Gy ward DDD (DDD/100) | ICU Ward DDD (DDD/100) |
|---|---|---|---|---|---|---|---|---|
| Amoxicillin | J01CA04 | Penicillin | 49 | 0.6 | 14 (0.2) | 21 (0.3) | 14 (0.2) | 0 |
| Amoxicillin + Clavulanic acid | J01CR02 | Beta-lactam-beta-lactamase inhibitors | 178.3 | 2.0 | 33.8(0.4) | 42.5 (0.5) | 42.08(0.5) | 60 (0.7) |
| Ampicillin | J01CA01 | Penicillin | 183.7 | 2.1 | 26.7(0.3) | 0.67 (0.01) | 152.33 (1.8) | 4 (0.05) |
| Azithromycin | J01FA10 | Macrolides | 70.8 | 0.8 | 62.75(0.7) | 0 | 3.5 (0.04) | 4.5 (0.05) |
| Benzathine penicillin | J01CE08 | Penicillin | 0.048 | 0.001 | 0.05(0.01) | 0 | 0 | 0 |
| Cephalexin | J01DB01 | First-generation cephalosporin | 182.5 | 2.11 | 35(0.4) | 36 (0.42) | 111.5 (1.3) | 0 |
| Cefepime | J01DE01 | Fourth-generation cephalosporin | 147 | 1.70 | 108 (1.3) | 2.5 (0.03) | 0 | 36.5(0.4) |
| Ceftazidime | J01DD02 | Third-generation cephalosporin | 1031.5 | 11.94 | 331 (3.83) | 146 (1.69) | 43.5 (0.5) | 511 (5.92) |
| Ceftriaxone | J01DD04 | Third-generation cephalosporin | 3567 | 41.29 | 1371.5(15.9) | 1171.5(13.6) | 563 (6.52) | 461 (5.34) |
| Ciprofloxacin | J01MA02 | Fluoroquinolone | 369.8 | 4.28 | 98 (1.13) | 169 (1.96) | 41.4 (0.48) | 61.4 (0.71) |
| Cloxacillin | J01CF02 | Penicillin | 32 | 0.37 | 20 (0.23) | 12 (0.14) | 0 | 0 |
| Cotrimoxazole | J01EE01 | Sulfonamides and trimethoprim | 38.88 | 0.45 | 32.64 (0.38) | 0 | 0 | 6.24 (0.07) |
| Doxycycline | J01AA02 | Tetracycline | 688 | 7.97 | 28 (0.3) | 0 | 652(7.6) | 8(0.1) |
| Gentamicin | J01GB03 | Aminoglycosides | 27.1 | 0.3 | 0 | 13.3(0.2) | 7.67 (0.1) | 6 (0.1) |
| Meropenem | J01DH02 | Carbepenem | 80.3 | 0.9 | 0 | 0 | 4.5 (0.05) | 75.8(0.9) |
| Metronidazole | J01XD01 | Imidazole | 1639.0 | 19.0 | 442.4(5.12) | 566.3(6.56) | 374(4.3) | 256.3(3.0) |
| Norfloxacin | J01MA06 | Fluoroquinolones | 28 | 0.3 | 4(0.1) | 21.6 (0.25) | 2.4(0.03) | 0 |
| Vancomycin | J01XA01 | Glycopeptides | 985 | 11.40 | 408.5(4.73) | 172.5 (2.00) | 25.5 (0.30) | 378.5(4.4) |
| **Total** | | | **9297.8** | **108.0** | **3016.3(35.0)** | **2375.0(27.5)** | **2037.4(23.6)** | **1869.3(21.7)** |

The study also showed that the average number of 1.3 antibiotics per patient, which is in line with reports ranging from 1.15 to 1.4 reported in Eritrea [22], Ethiopia [2], Zambia [25], and Pakistan [24]. However, it was lower than the average number of antibiotics prescribed per patient, according to WHO prescribing indicators, which range from 1.6 to 1.8 [25]. The lower the number of drugs prescribed per patient, the more appropriate the prescription practice, and it decreases disease problems caused by drug–drug interactions and adverse drug reactions.

The majority of antibiotics were prescribed for hospital-acquired infections (34.7%), followed by community-acquired infections (17.8%). The rest was for prophylaxis purposes. On the contrary, a study in Sierra Leone found that the most common indication for antibiotic use was community-acquired infection (51.9%), followed by surgical prophylaxis (23.8%) [26], and a study in Emergency Clinical County Hospital of Oradea, Romania, found that antibiotics were mostly prescribed for surgical and medical prophylaxes [27]. This indicates there might be a high burden of hospital-acquired infections in Ethiopia. Healthcare-associated infections and irrational use of antibiotics in healthcare settings are major global public health concerns and have bidirectional partnerships in the healthcare system. Where there is no transmission of infection, there is no need for antimicrobial treatment, thus reducing the development of resistance [28]. Therefore, enhancing infection prevention and control, and optimizing antimicrobial use at the national and facility level is very important.

The antibiotic consumption data were expressed as defined daily doses (DDD) per 100 bed day(BD). An overall antibiotic consumption for hospitalized patients in the current study was 108 DDD/100 BD. This finding was lower than the

**Table 7. Classes of antibiotics prescribed in adult medical, surgical, obstetrics/gynecology, and ICU wards of tertiary care hospitals in Ethiopia, 2022.**

| Serial number # | Class of antibiotics | # of antibiotics prescribed | % |
|---|---|---|---|
| 1. | Third-generation cephalosporin | 570 | 55 |
| 2. | Imidazole | 142 | 13.7 |
| 3. | Glycopeptides | 112 | 10.8 |
| 4. | Penicillins | 56 | 5.4 |
| 5. | Fluoroquinolones | 45 | 4.3 |
| 6. | Macrolides | 31 | 3 |
| 7. | Tetracycline | 19 | 1.8 |
| 8. | Fourth-generation cephalosporin | 18 | 1.7 |
| 9. | First-generation cephalosporin | 15 | 1.5 |
| 10. | Aminoglycoside | 10 | 2 |
| 11. | Carbepenem | 8 | 0.8 |
| 12. | Beta-lactam-beta-lactamase inhibitors | 6 | 0.6 |
| 13. | Sulfonamides and Trimethoprim | 3 | 0.3 |
| **Total** | | **1035** | **100.00** |

**Table 8. WHO AWaRe classification and number of antibiotics prescribed for adult patients in medical, surgical, obstetrics/gynecology, and ICU wards of tertiary care hospitals in Ethiopia, 2022.**

| AWaRe classes of antibiotics prescribed | Classes of antibiotics N (%) | # of antibiotics prescribed N (%) | DDD in gram | DDD/100BD (%) |
|---|---|---|---|---|
| **Access Group** | 10 (55.56) | 267 (26) | 3018.5 | 35 |
| **Watch Group** | 8 (44.44) | 768 (74) | 6279.3 | 73 |
| **Total** | **18 (100.00)** | **1035 (100)** | **9297.8** | **108** |

Note: N = number, AWaRe = Acess, Watch, Reserved.

antibiotic consumption in Nigeria, in which the total antibiotic consumption for hospitalized patients was 260.9 DDD/100 bed-days [29]. The three most frequently prescribed antibiotics were ceftriaxone (41.29 DDD/100 BD), Metronidazole (18.98 DDD/100 BD), and Ceftazidime (11.94 DDD/100 BD). This is in agreement with other studies, in Ethiopia in which ceftriaxone was the most frequently prescribed antibiotic [11]; in Sierra Leone hospitals, in which the most widely prescribed antibiotics were metronidazole and ceftriaxone [26], in Eritrea, Ceftriaxone was among the most commonly consumed antibiotics [7] and in Romania, the most prescribed antibiotics were ceftriaxone, followed by metronidazole, and cefuroxime [27]. However, it was inconsistent with a recent review of antibiotic use that reported penicillin is the most consumed (127.9 DDD/100 bed-days), followed by Cephalosporin (41.42 DDD/100 bed-days) and Fluoroquinolone (25.87 DDD/100 bed-days) [30].

As the consumption among wards of admission is considered, the medical ward had high annual antibiotics consumption (34.92 DDD/100BD), followed by surgical (27.5 DDD/100BD) and obstetrics/gynecology (23.59 DDD/100BD) wards. The ICU had low annual antibiotics consumption (21.64 DDD/100BD). On the contrary, a study conducted in Iran showed that the highest proportion of antibiotic consumption was observed in the obstetrics and gynecology wards, followed by surgical and internal medicine wards [22].

There were 18 types of antibiotics from 13 classes. Third-generation cephalosporins were the most prescribed antibiotics (55.12%), followed by imidazole (13.71%) and glycopeptides (10.81%). Similarly, a study in Iran found that

Cephalosporins were the most frequently prescribed antibiotics [22], and a study in the Emergency Clinical County Hospital, in Romania, reported that the most frequent were cephalosporins, 43.73% [27].

Regarding the WHO "AWaRe" classification of antibiotics prescribed, in our study, the "Watch" group (74.20%), antibiotics consumption was about 3 times greater than the "Access" group (25.80%). This widely deviated from the WHO recommendation that antibiotics in the "Access" group should account for at least 60% of total antibiotic consumption [31–33]. Antimicrobial resistance is one of the 21st century's greatest threats to health across the world [34]. The safest approach for the patients in different settings is to target increasing the proportion of consumption of Access antibiotics to at least 60% of total consumption [35]. Antibiotic pressure is particularly critical in hospitals, as hospitals treat numerous chronic illnesses and/or immunocompromised patients, and perform many invasive procedures, with a higher risk of healthcare-associated infections, dissemination of resistant micro-organisms, and severe outcomes [4]. The in-hospital use of antibiotics is an ever-growing concern and is attributed to a major proportion of the inflating drug budget, which poses a significant challenge for the health care services [36]. Thus, optimization of antibiotic utilization is very important to combat this critical distraction [29]. Hence, formulary restrictions, prescribers' education, advocating for hospital-specific formularies prioritizing "Access" group antibiotics, training on de-escalation protocols, and a strategy involving strengthening and integrating antimicrobial stewardship into health care facilities are critically needed in study settings.

Our finding was much higher than the findings in other studies; the "Watch" group accounts for 45.6% and the "Access" group accounts for 54.4%, in Northern Ethiopia [19] and the "Watch" group 51.7% and "Access" group 48.3% in Southwest Ethiopia [11]; of the total prescribed antibiotics, 42.1% were from "Watch" group, in Zambia [24], the "Watch" group was 54.3%, and "Access" group was 38.3% in Uganda [37], and a study in Romania, the use of Watch group antibiotics was 59.69% [27]. It was also slightly greater than a recent systematic review that reported over 70% of antibiotics were from the Watch category [16]. But similar to a study in Islamabad, in which 76% of antibiotics were prescribed from the "Watch" group [38].

Our findings clearly showed that there was a deviation of antibiotics consumption against the WHO recommendation for safe antibiotic use, which needs prompt intervention. Hence, interventions must be done to fix antibiotic consumption as well as strengthen antimicrobial stewardship in tertiary care hospitals to optimize antibiotic utilization in Ethiopia.

There was high consumption of antibiotics in the surgical ward (35.23%), followed by medical wards (28.09%), and the lowest in the ICU wards (15.64%). In agreement with our study, a study conducted in Sierra Leone found that there was the highest antibiotic consumption in the medical wards, followed by surgical wards [26].

## Limitations of the study

As we used the WHO DDD method to measure antibiotic consumption, this method can over- or underestimate antibiotic use, as it does not account for alternative dosing regimens and dose adjustment, which is unsuitable for pediatric settings since DDD is defined as the average dose in adults, and it may not reflect the dose used for a particular infection. This study also did not consider outpatients who were prescribed antibiotics. The findings from public tertiary hospitals may not reflect practices in primary and secondary care or the private sectors.

## Conclusion

There was a high consumption of antibiotics in tertiary hospitals in Ethiopia, as the majority of patients admitted received antibiotics. About three-fourths of antibiotics were prescribed from the "Watch" group, which was the inverse of the WHO recommendation. This has the potential to increase the burden of antibiotic resistance in the country. Only three classes of antibiotics, third-generation cephalosporins, imidazole, and glycopeptides, account for about four-fifths of antibiotics prescribed. The majority of antibiotics were prescribed for hospital-acquired infections. This study provides evidence for the necessity and a way forward for the establishment and revitalization of an antimicrobial stewardship in hospitals. Hence, interventions based on Ethiopia's national 'one health focused' AMR prevention and containment strategic plan,

specifically on enhancing infection prevention and control, and optimizing antimicrobial use, must be done for hospital-acquired infections and fix antibiotic consumption as well as strengthen antimicrobial stewardship in hospitals to optimize antibiotic utilization in Ethiopia. Future studies should expand methodology to include populations and integrate local anti-biogram to correlate prescribing with resistance trends, and we suggest multi-center studies including primary, general, and private hospitals for broader applicability.

## Supporting information

**S1 Data. Data of Antibiotics consumption.**
(XLSX)

## Acknowledgments

We would like to acknowledge Jimma University for providing the chance to study. Our gratitude extended to the data collectors and staff in all study tertiary care hospitals for their gratifying facilitation and support. Last but not least my gratitude going to my family especially my lovely wife for teacher Kumneger Dest and my kids Makida, Liyu and Kawone Abebe.

## Author contributions

**Conceptualization:** Abebe Kebede, Kifle Woldemichael, Sultan Suleman.

**Data curation:** Abebe Kebede.

**Formal analysis:** Abebe Kebede.

**Investigation:** Abebe Kebede.

**Methodology:** Abebe Kebede, Kifle Woldemichael, Alemseged Beyene Berha, Sultan Suleman.

**Project administration:** Abebe Kebede.

**Resources:** Abebe Kebede.

**Software:** Abebe Kebede.

**Supervision:** Abebe Kebede, Kifle Woldemichael, Sultan Suleman.

**Validation:** Abebe Kebede, Dereje Kebede, Kifle Woldemichael, Sileshi Dubale, Wendossen Shewarega, Alemseged Beyene Berha, Sultan Suleman.

**Visualization:** Abebe Kebede, Dereje Kebede, Kifle Woldemichael, Sileshi Dubale, Wendossen Shewarega, Alemseged Beyene Berha, Sultan Suleman.

**Writing – original draft:** Abebe Kebede.

**Writing – review & editing:** Abebe Kebede, Dereje Kebede, Kifle Woldemichael, Sileshi Dubale, Wendossen Shewarega, Alemseged Beyene Berha, Sultan Suleman.

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
