## [Decision Letter · Decision Letter 0]

31 Jul 2025

Dear Dr. Kebede,

Thank you for submitting your manuscript to PLOS ONE. After careful consideration, we feel that it has merit but does not fully meet PLOS ONE’s publication criteria as it currently stands. Therefore, we invite you to submit a revised version of the manuscript that addresses the points raised during the review process.

We look forward to receiving your revised manuscript.

Kind regards,

Tebelay Dilnessa, MSc

Academic Editor

PLOS ONE

3. Please update your submission to use the PLOS LaTeX template. The template and more information on our requirements for LaTeX submissions can be found at http://journals.plos.org/plosone/s/latex

Additional Editor Comments:

The paper requires intensive revision.

Avoid words like 'see figure 1' etc. It is better written as (Figure 1)

Follow the PLOS ONE manuscript writing protocol.

Reviewers' comments:

Reviewer's Responses to Questions

**Comments to the Author**

1. Is the manuscript technically sound, and do the data support the conclusions?

Reviewer #1: Partly

Reviewer #2: Partly

2. Has the statistical analysis been performed appropriately and rigorously?

Reviewer #1: Yes

Reviewer #2: Yes

3. Have the authors made all data underlying the findings in their manuscript fully available?

Reviewer #1: Yes

Reviewer #2: No

4. Is the manuscript presented in an intelligible fashion and written in standard English?

Reviewer #1: Yes

Reviewer #2: No

Reviewer #1: Dear Editors,

Thank you for inviting me to review the manuscript titled “Antibiotics Utilization Patterns among Tertiary Care Hospitals in Ethiopia.” The study addresses a critical public health issue and provides valuable insights for antimicrobial stewardship in low-resource settings. Below, I outline its strengths and key areas for improvement to enhance the manuscript’s rigor and impact.

Strengths:

• The manuscript is well-structured and contributes meaningfully to the literature on antibiotic use in sub-Saharan Africa.

• The application of WHO’s ATC/DDD methodology ensures standardized measurement of antibiotic consumption.

• Findings are actionable, particularly the overuse of "Watch" group antibiotics, which deviates from WHO recommendations.

Major Weaknesses and Recommendations:

1. Timeliness of Data:

The study was conducted three years ago, and antibiotic prescribing practices or resistance patterns may have since evolved. The authors should address how this temporal gap might affect the current relevance of their findings.

2. Ethical Approval/Consideration

While retrospective studies often involve anonymized data, the manuscript should explicitly state whether ethical approval (e.g., IRB reference number) was obtained to ensure compliance with research integrity standards.

3. Study Design Limitations:

o The retrospective cross-sectional design precludes causal inferences (e.g., linking prescribing patterns directly to resistance rates).

o Excluding outpatient and pediatric data omits significant antibiotic use contexts. Recommendation: Future studies should expand methodology to include these populations and integrate local antibiograms to correlate prescribing with resistance trends.

4. Generalizability Concerns:

o Findings from five public tertiary hospitals may not reflect practices in primary/secondary care or private sectors.

o Regional variability (4 regions) may not capture national heterogeneity. Recommendation: Acknowledge this limitation and suggest multi-center studies for broader applicability.

5. Methodological Gaps:

o The DDD metric does not account for dosing adjustments (e.g., renal impairment) or pediatric use.

o Clinical outcomes (e.g., treatment success, resistance emergence) are not analyzed. Recommendation: Discuss these limitations and propose linking future data to patient outcomes.

6. Presentation Issues:

o Tables 2–4 contain repetitive data and could be consolidated for clarity.

o A visual summary (e.g., bar chart of AWaRe group deviations from WHO targets) would improve readability.

7. Discussion Enhancements:

o Contextualize findings with regional AMR surveillance data (e.g., resistance rates for ceftriaxone).

o Compare results with Ethiopia’s AMR National Action Plan (if available) and propose specific stewardship interventions (e.g., audit/feedback, surgical prophylaxis guidelines).

8. Policy and Recommendations:

o Detail actionable strategies (e.g., formulary restrictions, prescriber education).

o Include cost-analysis of antibiotic misuse (e.g., budget impact of Watch-group overuse).

o Advocate for hospital-specific formularies prioritizing "Access" group antibiotics and training on de-escalation protocols.

Reviewer #2: Needs to work on English language presentation

What is the novelty of data in the study?

There is overall casual attitude in preparation of manuscript

Result: Table no 3, 5 and 6 are missing in the manuscript

Table no 2 mentions Drugs with drug name are mentioned as 1,2,3,4,5. This does not explain anything.

There is repetition of data presentation in the tables.

**Do you want your identity to be public for this peer review?** For information about this choice, including consent withdrawal, please see our Privacy Policy

Reviewer #1: **Yes: ** Bantayehu Addis Tegegne

Reviewer #2: No

---

## [Author Response · Author response to Decision Letter 1]

22 Aug 2025

Response to each point raised by the academic editor and reviewers were included in response to editor latter(Rebuttal later)

---

## [Decision Letter · Decision Letter 1]

1 Oct 2025

Dear Dr. Kebede,

Thank you for submitting your manuscript to PLOS ONE. After careful consideration, we feel that it has merit but does not fully meet PLOS ONE’s publication criteria as it currently stands. Therefore, we invite you to submit a revised version of the manuscript that addresses the points raised during the review process.

We look forward to receiving your revised manuscript.

Kind regards,

Tebelay Dilnessa, MSc

Academic Editor

PLOS ONE

Journal Requirements:

Additional Editor Comments:

It is better the author order, ‘Materials and methods’The figure 1, description should be placed within the manuscript where cited below it.Poofread is also required.

Reviewers' comments:

Reviewer's Responses to Questions

**Comments to the Author**

Reviewer #1: All comments have been addressed

Reviewer #2: (No Response)

2. Is the manuscript technically sound, and do the data support the conclusions?

Reviewer #1: Yes

Reviewer #2: Partly

3. Has the statistical analysis been performed appropriately and rigorously?

Reviewer #1: Yes

Reviewer #2: Yes

4. Have the authors made all data underlying the findings in their manuscript fully available?

Reviewer #1: Yes

Reviewer #2: Yes

5. Is the manuscript presented in an intelligible fashion and written in standard English?

Reviewer #1: Yes

Reviewer #2: No

Reviewer #1: Dear Editors,

Thank you for inviting me to review the manuscript titled “Antibiotics Utilization Patterns among Tertiary Care Hospitals in Ethiopia.” The study addresses a critical public health issue and provides valuable insights for antimicrobial stewardship in low-resource settings. Below, I outline its strengths and key areas for improvement to enhance the manuscript’s rigor and impact.

Strengths:

• The manuscript is well-structured and contributes meaningfully to the literature on antibiotic use in sub-Saharan Africa.

• The application of WHO’s ATC/DDD methodology ensures standardized measurement of antibiotic consumption.

• Findings are actionable, particularly the overuse of "Watch" group antibiotics, which deviates from WHO recommendations.

Major Weaknesses and Recommendations:

1. Timeliness of Data:

The study was conducted three years ago, and antibiotic prescribing practices or resistance patterns may have since evolved. The authors should address how this temporal gap might affect the current relevance of their findings.

2. Ethical Approval/Consideration

While retrospective studies often involve anonymized data, the manuscript should explicitly state whether ethical approval (e.g., IRB reference number) was obtained to ensure compliance with research integrity standards.

3. Study Design Limitations:

o The retrospective cross-sectional design precludes causal inferences (e.g., linking prescribing patterns directly to resistance rates).

o Excluding outpatient and pediatric data omits significant antibiotic use contexts. Recommendation: Future studies should expand methodology to include these populations and integrate local antibiograms to correlate prescribing with resistance trends.

4. Generalizability Concerns:

o Findings from five public tertiary hospitals may not reflect practices in primary/secondary care or private sectors.

o Regional variability (4 regions) may not capture national heterogeneity. Recommendation: Acknowledge this limitation and suggest multi-center studies for broader applicability.

5. Methodological Gaps:

o The DDD metric does not account for dosing adjustments (e.g., renal impairment) or pediatric use.

o Clinical outcomes (e.g., treatment success, resistance emergence) are not analyzed. Recommendation: Discuss these limitations and propose linking future data to patient outcomes.

6. Presentation Issues:

o Tables 2–4 contain repetitive data and could be consolidated for clarity.

o A visual summary (e.g., bar chart of AWaRe group deviations from WHO targets) would improve readability.

7. Discussion Enhancements:

o Contextualize findings with regional AMR surveillance data (e.g., resistance rates for ceftriaxone).

o Compare results with Ethiopia’s AMR National Action Plan (if available) and propose specific stewardship interventions (e.g., audit/feedback, surgical prophylaxis guidelines).

8. Policy and Recommendations:

o Detail actionable strategies (e.g., formulary restrictions, prescriber education).

o Include cost-analysis of antibiotic misuse (e.g., budget impact of Watch-group overuse).

o Advocate for hospital-specific formularies prioritizing "Access" group antibiotics and training on de-escalation protocols.

Reviewer #2: Inclusion criteria are vague and lack specification. Antibiotics outside ELM were not included?

Inclusion criteria mentions “Antibiotics prescribed for initial diagnosis were included”. Please clarify.

Ethics permission from the registered ethics committee of each hospital participating in the study is required to collect data. Verbal permission may not be sufficient or authentic.

Study design: The study design is mentioned as “retrospective cross-sectional study”; however, the exact time point of data collection is not mentioned in the manuscript. Only the duration during which the data was assessed, as has been mentioned

Results: The number of female participants in the text and table is different. Data on SD is missing. The legend for Table 2 is not clear. What are 1,2,3,4,5,6,7, rounds of prescribing? Table 4: column 1st: What does drug 1,2,3,4,56,7 indicate?

There are multiple instances of spelling errors.

**Do you want your identity to be public for this peer review?** For information about this choice, including consent withdrawal, please see our Privacy Policy

Reviewer #1: **Yes: ** Bantayehu Addis Tegegne, Lecturer of Pharmacology at Debre Markos University, Debre Markos, Ethiopia

Reviewer #2: No

---

## [Author Response · Author response to Decision Letter 2]

3 Oct 2025

All comments of reviewers were responded and attached as response to reviewers and in main document

---

## [Editor Report · Decision Letter 2]

6 Oct 2025

Antibiotics Utilization Patterns among Tertiary Care Hospitals in Ethiopia

PONE-D-25-34538R2

Dear Dr. Kebede,

We’re pleased to inform you that your manuscript has been judged scientifically suitable for publication and will be formally accepted for publication once it meets all outstanding technical requirements.

Kind regards,

Tebelay Dilnessa, MSc

Academic Editor

PLOS ONE
---

## [Editor Report · Acceptance letter]

PONE-D-25-34538R2

PLOS ONE

Dear Dr. Kebede,

I'm pleased to inform you that your manuscript has been deemed suitable for publication in PLOS ONE. Congratulations! Your manuscript is now being handed over to our production team.

Kind regards,

on behalf of

Dr. Tebelay Dilnessa

Academic Editor

PLOS ONE